# Systematic Review: Fragile X Syndrome Across the Lifespan with a Focus on Genetics, Neurodevelopmental, Behavioral and Psychiatric Associations

**DOI:** 10.3390/genes16020149

**Published:** 2025-01-25

**Authors:** Ann C. Genovese, Merlin G. Butler

**Affiliations:** Department of Psychiatry and Behavioral Sciences, University of Kansas Medical Center, 3901 Rainbow Blvd., MS 4015, Kansas City, KS 66160, USA; mbutler4@kumc.edu

**Keywords:** fragile X syndrome, lifespan, genetics, neurodevelopmental, behavioral, psychiatric

## Abstract

Background/Objectives: Fragile X syndrome (FXS) is one of the most common genetic causes of intellectual developmental disability and autism spectrum disorder (ASD), second only to Down’s syndrome and associated with a broad range of neurodevelopmental, behavioral, and psychiatric challenges. FXS may be present in infants or young children with characteristic dysmorphic features, developmental delays, and behavioral challenges. The diagnosis of FXS is confirmed by the molecular genetic testing of the *FMR1* gene encoding fragile X messenger RNA-binding protein (FMRP), involved in regulating the translation of multiple mRNAs which play a key role in neuronal development and synaptic plasticity. Understanding the genetic cause, pathophysiology, and natural history of FXS is crucial for identifying commonly associated comorbidities, instituting effective therapeutic interventions, and improving long-term outcomes. Methods: This systematic review employed a comprehensive literature search using multiple electronic databases including PubMed, Web of Science, and Scopus with keywords related to fragile X syndrome, lifespan, genetics, neurodevelopmental, behavioral, and psychiatric disorders. Results: FXS is associated with an increased risk for specific neurodevelopmental, or psychiatric disorders. Symptoms and challenges associated with FXS vary based on multiple factors, including genetic differences, age, sex, comorbid conditions, various environmental influences, the availability of support, and opportunities for therapeutic interventions. Knowledge of these associations helps guide caregivers and clinicians in identifying potentially treatable conditions that can help to improve the lives of affected patients and their families. Conclusions: The focus of this article is to explore and describe the genetic underpinnings of FXS, identify associated developmental, behavioral, and psychiatric conditions over the lifespan, and provide a review of clinical features, therapeutic interventions including investigational treatments, and current research updates.

## 1. Introduction

Fragile X syndrome (FXS) (OMIM# 300624) has substantial impacts on individuals across the lifespan, affecting both males and females, with characteristic signs and symptoms which often indicate the presence of the disorder present from a very young age (OMIM-Online Inheritance in Man, www.omin.org [1]). FXS in early childhood is generally characterized by developmental delays, repetitive behaviors, impaired social interactions, and difficulty with emotional self-regulation. Behavioral phenotypes, described as having a greater probability of exhibiting specific behavioral patterns related to the syndrome, have been observed with FXS, although there is variability in behavioral features [2]. The characteristic behavioral phenotype of FXS is relatively consistent, with gaze avoidance, anxiety, hyperactivity, distractibility, emotional lability, stereotypic movements, and echolalia [3]. The mean age of FXS diagnosis is 36 months in the United States, as developmental delays can be subtle in infancy, with language delays generally becoming more obvious by 2 to 3 years of age [4].

During the first year of life, delays are not usually detected, but hypotonia, and feeding difficulties with the gastrointestinal reflex and vomiting are common [5,6]. By 24 months, language delays are apparent, generally followed by irritability and tantrums by the age of 3 years [7]. Although children with FXS often do not present with characteristic dysmorphic features, there are a range of findings that change over the lifespan, such as Prader–Willi-like phenotype in a small proportion of FXS males with onset hyperphagia and severe obesity, hypogonadism, and delayed puberty [8,9]. Other findings include reproductive issues such as premature menopause in FXS females, depression, and fragile X-related ataxia with tremors, appearing similar to those in Parkinson’s disorder [10,11].

School-aged children and adolescents with FXS typically face challenges with learning and academic performance, with deficits in reciprocal communication, verbal comprehension, and understanding of complex concepts, often benefiting from evidence based individualized educational interventions including specific classroom accommodations, or special education support [12]. In addition, developmental, behavioral, or psychiatric disorders occur more commonly than in their typically developing peers [13]. The long-term prognosis in FXS is generally improved with early diagnosis and intervention services including social skills training, life skills instruction, vocational support, and mental health services when indicated, which can significantly improve functional abilities and quality of life [14].

Individuals with FXS may experience difficulties in navigating transitions or adapting to changes in their daily lives, particularly those which occur following milestones such as high school graduation, and often experience challenges in achieving and maintaining personal, social, and financial independence in adulthood. Developmental delays often persist in adulthood, affecting psychosocial functioning, educational attainment, and employment opportunities. Although there are generally limitations in independent living skills and employment capabilities, some adults with FXS achieve significant independence when provided with appropriate preparation, instruction, training, support, and workplace accommodations [15].

Many variables impact the clinical course and outcomes for individuals with FXS. There can be significant variability in the severity of FXS-associated impairments and the risk for co-occurring conditions, even within families. Optimal outcomes in adult life for individuals with developmental disabilities are determined by many factors, including acquiri8ng of independent living skills, gaining proficiency in activities of daily living, developing support networks, maintaining friendships, obtaining employment, and participating in leisure activities. For men with FXS, the level of functional skills is the strongest predictor of achieving independence in adult life, whereas for women, the ability to interact appropriately in social situations is the strongest predictor [16].

Co-occurring developmental or mental health conditions influence independence in adult life, most commonly autism spectrum disorder (ASD) for men and mood disorders for women [15]. For those with behavior problems or recurrent self-injury associated with FXS, there are scant data to help predict how these concerns tend to progress as individuals mature, and particularly following transition from adolescence into adulthood [17,18]. The identification and treatment of mental health conditions, most commonly anxiety or depressive disorders, improve long-term prognosis in terms of functional outcomes and quality of life [19]. For those with both FXS and ASD, there is evidence that autism severity tends to decrease with maturation [20].

FXS has significant effects on functional abilities throughout life but does not generally progress or worsen significantly over time. A normal life span is expected in FXS with no life-threatening health concerns directly related to the syndrome itself. Adults with FXS experience a range of outcomes depending on the severity of impairments in cognitive, communication and social skills. Receptive language abilities may be one of the strongest predictors of independent daily living skills performance [21]. There have been no large longitudinal studies that assess the molecular variations and behavior or cognitive correlations. Additional longitudinal studies are necessary to assess the developmental trajectories of FXS across the lifetime and relate the outcomes to molecular and environmental factors [22].

## 2. Genetics of Fragile X Syndrome

The first X- linked pedigree of intellectual disability reported by Martin and Bell in 1949 was later identified to be FXS when analyzed with the *FMR1* gene DNA repeat expansion test after the discovery of the *FMR1* gene in 1991 [23,24]. Approximately 2–6% of all individuals diagnosed with ASD have FXS, the vast majority of which are males, resulting in the consensus that all people diagnosed with ASD should undergo genetic testing for FXS [25]. FXS affects approximately 1 in 4000 males and 1 in 8000 females, with males typically exhibiting more severe symptoms due to their single X chromosome, as the second X chromosome in females helps to compensate for the affected X chromosome [26].

Historically, screening and diagnostic techniques for FXS can include prenatal testing, specifically amniocentesis and chorionic villus sampling, recommended for families with a history of FXS or related symptoms, or postnatal testing such as polymerase chain reaction (PCR) combined with Southern blot analysis and triplet-primed PCR (TP-PCR) for identifying *FMR1* gene mutations. Current standards outline the use of PCR and methylation-sensitive techniques for a comprehensive diagnostic evaluation and genetic counseling for family planning [27]. Early detection of FXS with methylation specific-quantitative melt analysis (MS-QMA) that targets CpG sites within the *FMR1* intron 1 has been achieved using the DNA of newborn blood spots from birth in both sexes [28].

The *FMR1* gene located on the X chromosome was initially identified as a fragile site at Xq27.3 and identified as the genetic cause of FXS [29,30]. The *FMR1* gene encodes the fragile X RNA-binding protein (FMRP), which is highly expressed in the brain and testicular tissue, but also affects other organ systems, playing an essential role in a variety of biological processes [31]. The normal allele of the *FMR1* gene typically has 5 to 40 CGG repeats in the 5′ untranslated region. Repeat expansion within this region is the most common cause of FXS. On rare occasions, other *FMR1* gene variants are found in patients with FXS involving the coding sequence and not the CGG repeat region, as summarized by Sitzman et al. [32].

FXS is inherited in an X-linked dominant pattern, with mothers being carriers of the altered gene, passing it on to both male and female offspring. Abnormal alleles of dynamic mutations are reported to include the fragile X premutation (55–200 CGG repeats) and the full mutation (>200 CGG repeats), which renders the *FMR1* gene non-functional. These *FMR1* gene changes are dynamic and can expand between generations, particularly when passed from a female to her children, with the potential of expanding from a premutation to a full mutation causing FXS. The *FMR1* gene CGG repeats between 55 and 200 define a premutation or carrier status and are estimated to occur in about 1 in 250 females and about 1 in 800 males in the general population [33].

The most common *FMR1* gene mutation is due to an expansion of the CGG trinucleotide repeat region, with greater repeat expansion leading to more clinical severity. This expansion leads to the gene methylation of the cytosine bases and silences *FMR1* gene activity with significantly reduced or no production of mRNA and encoded FMRP production. FMRP acts as a regulatory protein that binds to specific mRNAs and controls their translation into neuronal proteins at the synapse and connective tissue, including collagen [34,35]. This regulation is crucial in the development of synaptic plasticity, which creates the ability of synapses to strengthen or diminish over time, a fundamental mechanism underlying learning and memory [36]. Essential for normal cognitive development, FMRP regulates nearly all aspects of neural progenitor cell proliferation and differentiation. The absence of FMRP disrupts neuronal function at the synapse level, thus negatively impacting interneuron connectivity, neural circuit maturation and synaptic plasticity, along with connective tissue involvement, ultimately resulting in joint laxity, tissue fragility, hernias and mitral valve prolapse [35].

To further understand the role of the *FMR1* gene and its protein production, the STRING database [37] was used to identify predicted protein–protein associations, functional interactions, and biological network analysis [38]. This analysis found ten nodes for the FMR1 protein (FMRP). Each node represents the proteins produced, including isoforms and 44 edges, which indicate both direct and predicted functional and physical protein–protein associations with interactions for each gene (see Figure 1). Functional enrichments related to the *FMR1* gene involved biological processes, molecular functions, cellular components, pathways, and disease–gene associations, as shown in Table 1.

The top ten significantly associated proteins or predicted functional partners found for the *FMR1* gene using the STRING database are depicted in Figure 1. These include CYFIP2, CYFIP1, AGO2, FXR2, NUFIP2, FXR1, DICER1, PURA, AGO1 and EIF4E. The proposed functions and description of these ten genes with their encoded proteins with interactions are shown in Table 2 and lend evidence for synaptic functional regulators with multifunctional polyribosome-associated RNA binding. The *FMR1* gene and its encoded protein (FMRP) plays a central role in neuronal development and synaptic plasticity via alternative mRNA stability in neurons, dendritic transport, and postsynaptic local protein synthesis, along with connective tissue protein supported by other sources [39,40,41].

The information accessed on the STRING website is shown in Table 1, and descriptions of the top ten recognized proteins and their functions interacting with FMRP are shown in Table 2. These proteins are encoded by ten interactive genes with *FMR1*, sharing the common processes and functions listed when disturbed and lead to FXS with multi-organ system involvement, specifically neurodevelopment and synaptic plasticity with an impact on multiple mRNA transcription. Hence, FMR1-encoded protein or FMRP acts as a synaptic functional regulator with multi-functional polyribosome-associated RNA-binding protein and properties, playing a central role in neuronal development and synaptic plasticity. This occurs through the regulation of alternative mRNA splicing, stability and nuclear export involved in dendritic transport and postsynaptic local protein synthesis via a subset of messenger RNAs. The factors contributing to the FXS neurodevelopmental, behavioral, and clinical presentation include neuronal cell division, regulation and apoptosis, allowing for specific cell division and tissue types, as well as their growth, loss and timely reorganization for normal organ development and function, specifically neurodevelopment [37].

Errors in processing are recognized in FXS through an enlarged head size with potential axon overgrowth and through the timing of craniofacial and brain malformations and dysfunction. The precise timing, quantity, and diversity of specific mRNA from related genes are critical for normal transcription, repression, and encoded protein translation within critical embryonic intervals that are key to normal neurodevelopment and function. Hence, the pathophysiological link between the genetic cause and resultant clinical manifestations in FXS is explained by the role of FMRP in regulating the translation of mRNAs involved in many critical brain pathways. These include glutamate receptor signaling, synaptic long-term depression and potentiation pathways, neurotransmitter GABA receptor signaling, cAMP/cGMP signaling, and the regulation of RAC1 (RAS-related C3 botulinum toxin substrate 1), producing a variety of functions, including phagocytosis, mesenchymal-like migration, modulation of cytoskeleton, neuronal polarization, and axonal growth. The pathways also involve circadian expression [42,43,44] and RAC1 interactions with CYFIP1 and FMRP [37].

Animal models of FXS, including invertebrate (Drosophila) and vertebrate (mouse, rat, zebrafish), have focused on loss-of-function models with the disruption or knockout (KO) of the *FMR1* gene homolog [45,46,47,48]. These models have provided insights into the neuropathological effects resulting from a loss of an FMRP non-human equivalent and have expanded our understanding of the learning, motor, cognitive, and behavioral deficits associated with FXS. However, the findings have been inconsistent, as no single animal model has been able to fully recreate the FXS phenotype [49,50,51].

## 3. Phenotypic and General Health Features

The phenotype and dysmorphology of FXS are related to a deficit in the FMR protein (FMRP), and this is typically manifested by a long narrow face, large, wide and prominent ears, a high arched palate, and features of a connective tissue disorder, such as hyperextensible finger joints, pectus excavatum, flat feet, hernias, soft skin and mitral valve prolapse [35], as well as low muscle tone and pubertal macro-orchidism (enlarged testicles) in males [52,53]. Up to 30% of young children with FXS will not have dysmorphic features, and about 5–10% of children with FXS display a Prader–Willi-like phenotype, including severe obesity, hyperphagia, hypogonadism, behavioral problems, and delayed puberty [8,9,54,55]. Neural imaging studies have shown ventriculomegaly (40% of cases), along with hippocampal and caudate nucleus hypertrophy with a small vermis [56]. Seizures are observed in about one-fourth of individuals with FXS and are typically complex partial and generalized tonic–clonic but in most cases can be well controlled with anti-convulsant medications [57].

A mild increase in head size is generally noted in FXS, and growth is often above the 50th percentile in prepubertal males; however, about one-fourth of adults with FXS are below the fifth percentile for height [58,59]. A high prevalence of recurrent otitis media with associated hearing loss is found in 63% of young FXS males compared with 38% of children with developmental disabilities but without FXS, and in 15% of aged-matched controls [60]. Strabismus is observed in 30 to 55% of young males with FXS and should be evaluated before four years of age by an ophthalmologist [61].

Connective tissue problems are noted in FXS and include flat feet, scoliosis, pectus excavatum and joint laxity, but joint dislocations generally do not occur [35,62]. Soft skin, mild cutis laxa, a prominent forehead with an elongated face, prognathism after puberty, hernias and healing concerns are also common [5]. Connective tissue abnormalities may also play a role in macro-orchidism, seen primarily in adults with FXS. Mitral valve prolapse is uncommon in childhood but present in about one-half of adults with FXS [63].

Health outcomes in FXS may be associated with variations in the *FMR1* gene, with related FMRP levels and epigenetic changes impacting clinical features, including intentional tremors, gait ataxia, parkinsonism, neuropathy, the autonomic nervous system, and reproduction [11,34,64]. Fragile X premutation carriers also experience health impacts. The fragile X premutation is associated with an elevated risk of developing either fragile X-associated primary ovarian insufficiency (FXPOI, OMIM# 311360), resulting in reproductive issues and premature menopause [65], fragile X-associated tremor/ataxia syndrome (FXTAS, OMIM# 300623), a progressive neurodegenerative disorder occurring later in life [34], or fragile X-associated neuropsychiatric disorders (FXANDs), most often neurodevelopmental, anxiety or depressive disorders [66].

Altered sleep patterns and dysregulated melatonin profiles have been demonstrated in boys with FXS when compared with age-matched normal controls. Sleep disturbances, including sleep onset insomnia or interrupted sleep, are common in FXS and can have several impacts [67]. The sleep architecture in boys with FXS shows significant elevations in nocturnal melatonin production, greater variability in total sleep time, more difficulty with sleep maintenance, and the impaired development of language skills [68,69]. Obstructive sleep apnea, particularly in adolescent boys, has also been more commonly reported in FXS, and given the correlation between sleep apnea and irritability, it is particularly important to consider this possibility, especially in the context of loud snoring [70].

## 4. Neurodevelopmental Features

Infants and toddlers with FXS often exhibit developmental delays. which can be manifested by impairments in motor coordination, gross and fine motor skills, speech acquisition, language development, reciprocal communication, and social skills. Both boys and girls with FXS will demonstrate global delays in affecting most of the expected early childhood developmental milestones by age 2, with boys showing delays at as young as 6 months old, compared with girls, who typically demonstrate these same delays at about 1 year of age. It should be noted that in most cases, these developmental delays are identified much earlier than the child’s diagnosis of FXS [71].

ASD and FXS are genetic neurodevelopmental disorders with different but potentially related neurobiological underpinnings, which explains the significant overlap in their behavioral features. Just over 4 in 10 people with FXS have comorbid ASD, including 46% of males and 16% of females with FXS [72]. Although FXS is a genetic disorder of a known cause, ASD is a complex genetic disorder, with its causation likely being related to both genetic and environmental interaction effects [73]. Milder ASD-like features including social anxiety, extreme shyness, eye gaze avoidance and sensory aversions are common in FXS, even in those not meeting the full criteria for a formal autism diagnosis [74].

Intellectual developmental disability (IDD) or other learning disorders are present in most individuals with FXS, with associated functional impairment ranging from mild to severe. Cognitive difficulties impact learning, comprehension, problem solving, working memory, executive function, and language processing abilities. The deficiency of FMRP in the full mutation nearly always leads to intellectual deficits in males with FXS [75]. In contrast, mosaicism, which involves the presence of cells with either methylated full mutation or smaller unmethylated pre-mutation (CGG 55–199) alleles in the same individual, have been associated with better cognitive functioning [76].

Males with fragile X syndrome generally demonstrate a decline in IQ scores over their lifespan, with the most marked declines seen during the early pubertal period, suggesting slowing neuronal development [77]. Age-related cognitive decline often starts earlier for men with FXS than for men in the general population, in many cases beginning before middle age, usually without any medical or other underlying cause [78]. In females with FXS, however, only 25% will have an intelligence quotient (IQ) below 70 (with an additional 50% of women with an IQ in the borderline range of intellectual functioning), due to the encoded production of FMRP by the unaffected X chromosome [8].

Despite the weaknesses in global cognitive development, there are areas in which individuals with FXS display relative strengths, including simultaneous informational processing and long-term memory. The speech intelligibility of individuals with FXS is usually good, and the decoding of words is better than expected based on nonverbal cognitive abilities. In contrast to individuals with autism, individuals with FXS generally demonstrate strengths in conversational turn-taking activity. Although language development is delayed, the majority of individuals with FXS become oral communicators, albeit with relatively stronger receptive language capabilities in comparison to expressive language capabilities [79].

## 5. Neurological Features

CNS structural, functional connectivity and electroencephalographic differences have been demonstrated in FXS, including converging structural and functional abnormalities in the left insular cortex, a region implicated in ASD. This suggests compromises in insula integrity and connectivity, which could prove useful in establishing an imaging biomarker for FXS [80]. Anatomical aberrations occur early in development and endure throughout adulthood, with caudate enlargement, a smaller left superior temporal gyrus, and smaller insula being among the most consistently reported findings and have been demonstrated in 1- to 3-year-old boys, adults, and, most recently, in adolescents with FXS [81,82,83].

FXS and ASD have been described as disorders of neural connectivity, with impacts including an imbalance in the ratio of excitatory to inhibitory synaptic transmission and altered synaptic structure and function [84,85]. Structural connectivity, referring to the physical connections of synapses, neuronal networks, and tracts, as well as the functional connectivity of the central nervous system, referring to the temporal correlations between spatially distinct regions, are disrupted in both FXS and ASD [86]. When evaluated, about three-quarters of young children with FXS have been found to have electroencephalogram (EEG) abnormalities, such as focal spikes; however, about one-third of these will experience a spontaneous remission of EEG abnormalities by mid-childhood [87].

Seizure disorders occur often in FXS, as seen in most other neurodevelopmental disorders [88]. Approximately 20% of males and 6% of females with FXS are reported to have seizures, and those affected tend to have more severe developmental and behavioral problems including ASD [89]. Focal or localized seizures, generalized seizures, or both occur in patients with FXS, with focal or localized seizures being the most common type. The mean age of seizure onset in FXS is about 6 years of age, with the great majority having an onset of seizures prior to age 10. In most cases, seizures in FXS can be well controlled with medications and tend to resolve during childhood [90,91]. Seizures may add to the severity of the phenotype because animal studies have shown that FMRP is released by the dendrites during seizures, thereby depleting the dendrites of their regulatory effects [92].

Abnormal sensory processing, including auditory hypersensitivity, impaired habituation to repeated sounds, reduced auditory attention, and visuospatial impairments, have been documented in studies of infants and toddlers with FXS [93]. Children with FXS have elevated reactivity to audiogenic sensory stimuli, demonstrated by enhanced amplitudes in electrodermal studies, as well as a lack of habituation to repetitive stimuli [94]. Electroencephalograms (EEGs) and magnetoencephalogram (MEG) studies have also demonstrated enhanced electromagnetic responses to auditory stimuli, indicating a heightened neural response to sound in the brain [54]. Also notable is that abnormal responses of the auditory cortex to sound that are present from early development may have a negative impact on the development of communication and language skills [95,96].

## 6. Behavioral and Psychiatric Features

Behavioral challenges are a hallmark of FXS, often manifested by difficulty maintaining eye contact, excessive shyness, social withdrawal, limited or awkward social skills, and are often associated with or made worse by underlying emotional dysregulation or sensory issues. Rigid, repetitive, or stereotypical behaviors such as hand-flapping or rocking, repetitive or perseverative speech, or obsessive and compulsive-type behaviors are also common aspects of FXS [13]. These behaviors typically cause significant impairments in daily functioning and can persist throughout life, which makes early identification and intervention critical [97]. Behavioral and social problems are not only distressing for families, other caregivers, educators, and peers, but can be obstacles to successful integration within the greater society [98].

Excessive irritability, temper tantrums, aggression toward others, property destruction or self-injurious behaviors (such as head-banging, hitting, slapping, or hand biting) occur in a subset of individuals with FXS, and can pose a serious safety risk [99]. These behaviors can be exacerbated by environmental factors, such as physical discomfort, excessive sensory stimuli, external demands, frustrations, non-preferred activities, changes in routine, or other stressors [100]. These behavior problems can occur at any age but are more common in males, with more than 40% of males being diagnosed with or treated for problems with aggression and self-injury, compared to less than half that rate in females [15]. Elevated levels of anxiety, irritability, attention, impulsivity, and hyperactivity often cause or contribute to other behavior problems often associated with FXS [101].

The most common psychiatric disorders associated with FXS are attention deficit hyperactivity disorder (ADHD), anxiety disorders, and affective (mood) disorders [13]. Compared with controls, boys with FXS exhibit significantly lower levels of attentional focus, inhibitory control, and soothability, with higher activity levels and a greater tendency to seek intense activities. The cardinal symptoms of ADHD include distractibility, hyperactivity, and impulsivity, all of which are more frequent in individuals of all ages with FXS, and when significantly interfering, may indicate a diagnosis of ADHD [102]. With about 80% of males and 30% of females with FXS meeting the criteria for ADHD, many regard ADHD-related deficits as part of the core FXS behavioral phenotype [103,104].

Depression and anxiety have reportedly been found to occur in about one-half to more than two-thirds of males and females with FXS and can occur across the lifespan [15]. Approximately 70% of individuals with FXS have at least one anxiety disorder compared to about 10% of age-matched individuals in the general population. With their onset typically in childhood, generalized anxiety disorder, specific phobias, social anxiety disorder, and obsessive–compulsive disorder are the most common psychiatric disorders associated with FXS [105]. In fact, social anxiety is considered a hallmark feature of FXS, often manifested by the so-called “Fragile X handshake”, where the individuals may shake the interviewer’s hand but will avoid eye contact until the interviewer looks away [106].

Affective disorders which include major depression (with or without psychotic features) or persistent depressive disorder should be suspected when there are unexpected changes in mood, increased irritability, social isolation, loss of interest in usual activities, suicidal thoughts, fatigue, difficulty concentrating, appetite changes, or sleep disturbances. Major depression is more likely to occur in association with neurodevelopmental disorders including FXS but is even more prevalent in the fragile X premutation [13]. Depressive disorders occur in about 40% of premutation carriers overall, but in about 65% of carriers who are later diagnosed with FXTAS. Given that depression often manifests prior to the onset of motor (tremor and ataxia) symptoms, mood changes could represent a predictor of premutation carriers who are at risk of developing FXTAS [66].

## 7. Treatment Interventions

Speech language therapy instituted in the early developmental years is critical in addressing delayed language and communication skills in FXS, with intervention goals of improving receptive and expressive vocabulary, morphology, syntax, pragmatic skills, narrative language, and literacy [41,79]. Greater caregiver engagement and responsiveness, as well as consistent use of language facilitation strategies, are highly predictive of later reciprocal communication ability in FXS [107,108]. With more complex communication impairments, and particularly in the case of non-speaking children, the use of augmentative and alternative communication strategies or devices, including communication boards, picture symbols, sign language, and speech-generating devices, can also enhance language and communication skills [109].

Challenging behaviors including aggression, self-injury, and property destruction, which occur in about 60 to 80% of boys with FXS, can cause significant distress and interfere with the child’s educational and social development [10]. As with all developmental disorders, early behavioral interventions starting at a young age are considered crucial in improving long-term outcomes. Applied behavior analysis [110] or a similar behavior-based therapy, based on the principles of behavior modification and learning theory, is often recommended to address developmental and behavioral challenges, particularly for children with FXS and ASD. However, the benefit of these intervention practices in individuals with FXS alone is less clear, with evidence suggesting that implementation strategies may need to be tailored to be effective with the FXS behavioral phenotype [111].

Behavioral and parent training interventions are often beneficial in reducing aggression, hyperactivity, and tantrums in children and adolescents with FXS [112]. Specific evidence-based parent-mediated interventions including functional communication training, cooperative parent-mediated therapy, and parent–child interaction therapy have been demonstrated to provide durable positive effects in boys with FXS [113,114,115,116]. In higher-functioning individuals with FXS, individual psychotherapy, including cognitive behavior therapy (CBT), can improve anxiety, depression, and social communication skills [117]. Services for individuals of all ages with fragile X syndrome should not only target functional and interpersonal skills but also co-occurring mental health conditions to support optimal outcomes [15].

There are no FDA-approved medications for FXS; however, there are several interventions and treatments aimed at managing the symptoms and improving the quality of life of individuals with FXS [118]. Symptom-based treatments, including psychotropic medications, can help improve attention and learning or reduce distressing symptoms. Stimulant or nonstimulant medications are commonly prescribed for symptoms of hyperactivity, impulsivity, and inattention, while SSRIs (selective serotonin reuptake inhibitors) or SNRIs (serotonin norepinephrine reuptake inhibitors) may help with anxiety and mood disorders. Antipsychotics or other mood stabilizers, including anticonvulsants, may help to reduce persistent irritability, tantrums, recurrent self-injurious behaviors, or impulsive aggression [13].

It is recognized that FXS has no cure. There are no genetic manipulations, medical interventions, or pharmacological agents that have been shown to reverse the full impact of the lack of FMRP on the developing brain during fetal development [119]. Pharmacological interventions for core behavioral features of FXS which have been investigated in clinical trials to date have failed to demonstrate consistent therapeutic benefits [120]. Currently, several available medications or supplements, such as metformin, sertraline, and cannabidiol (CBD), have demonstrated modest benefits and are frequently used off-label by clinicians to treat either the core behavioral manifestations in FXS or to target the commonly associated symptoms such as irritability or anxiety. A combination of developmental supports and behavioral therapy, with the addition of psychotherapy and/or targeted, evidence-based pharmacotherapy when clinically indicated, is currently considered the most effective intervention strategy for individuals with FXS [41].

## 8. Investigational Treatments

Ongoing clinical trials and emerging therapies in FXS currently being investigated may improve cognitive and behavioral outcomes by targeting the FMR1 gene pathway and other related mechanisms. Animal studies, which have been robust in this genetic disorder, have contributed a wealth of knowledge regarding understanding the molecular, cellular, physiological, and behavioral impairments associated with FXS; however, no single animal model has been able to fully recreate the human FXS phenotype [121]. Human clinical trials research has focused on various therapeutic candidates including mGluR5 antagonists, GABA-B agonists, Phosphodiesterase-4D (PDE4D) inhibitors and other drugs aimed at modulating the underlying mechanisms disrupted by FMRP deficiency.

Metformin is a commonly prescribed antidiabetic drug which has been shown to reduce core deficits in animal models of FXS, reducing many associated impairments and anomalies including social skills deficits, repetitive behaviors, macroorchidism, aberrant dendritic spine morphology, and a decrease in synaptic transmission in the adult FXS mouse model [122]. Human clinical trials have shown that metformin administration can result in positive behavioral changes, with reductions in social avoidance, irritability and aggression, enhanced cognitive functioning, the stabilization of IQ scores and improvements in adaptive behaviors over time, which are particularly promising results given that there is typically an age-related deterioration of these skills in FXS [123,124].

Cannabidiol has shown promising results in recent pharmaceutical studies in FXS. Multiple lines of evidence suggest a central role for the endocannabinoid system (ECS) in the pathogenesis of FXS [125]. The absence of FMRP in FXS disrupts ECS signaling, which has been implicated in FXS pathogenesis. The ECS facilitates synaptic homeostasis and plasticity through the cannabinoid receptor 1 (CB_1_). Cannabidiol may help restore synaptic homeostasis by acting as a negative allosteric modulator of CB_1_, affecting DNA methylation, serotonin 5HT_1A_ signal transduction, and gamma-aminobutyric acid (GABA) and dopamine D_2_ and D_3_ receptor signaling, which may contribute to beneficial effects in FXS. A transdermal cannabidiol gel *ZYN002*, manufactured as a synthetic form of CBD, has been shown to lessen behavioral abnormalities, including social avoidance, irritability, hyperactivity, and inappropriate speech in children with FXS [126].

Phosphodiesterase-4D (PDE4D) inhibitors, which have demonstrated potential for the treatment of various neurodegenerative diseases including Alzheimer’s dementia, are also being investigated to determine their ability to lead to an improvement in cognitive function and behavioral outcomes in FXS [127]. PDE4D is a key modulator of cAMP levels, which are relevant to learning and memory. Reduced levels of cAMP in FXS have been confirmed in multiple FXS human cell line studies [128,129,130]. Both FXS *Drosophila* and FXS mouse model studies have decreased cAMP in the brain and have shown behavioral deficits reversed by genetic or pharmacological manipulations that restore cAMP levels [131,132,133]. BPN14770 is a PDE4D allosteric inhibitor which modulates cAMP and in turn promotes the maturation of synaptic connections between CNS neurons. In a Phase 2 trial of 30 adult males with FXS, BPN14770 was associated with improved cognition, language skills and daily functioning [134]. Phase 3 trials with open-label extensions are currently ongoing to confirm its benefits in a larger cohort and to determine whether these benefits are sustained over time.

Personalized medicine approaches, particularly genetic counseling, pharmacogenetic testing and targeted medical treatment interventions, are becoming the standard of care in the evaluation and treatment of individuals with FXS. Recent studies suggest that restoring the function of FMRP or targeting specific molecular pathways could potentially alleviate cognitive and behavioral symptoms of FXS. Investigations are ongoing to determine whether gene editing can be used to demethylate the *FMR1* gene promoter region to improve patient outcomes. Furthermore, clustered regularly interspaced palindromic repeats (CRISPR)/Cas9 and developed nuclease defective Cas9 (dCas9) strategies may be promising options for genome editing in gain-of-function mutations by rewriting new genetic information into a specified DNA site [135].

## 9. Limitations

Noted limitations for this review include a relative lack of data on neurodevelopmental, behavioral, and psychiatric comorbidities in individuals with FXS, especially for those with the dual diagnosis of FXS and ASD. The scarce availability of resources such as genomic technology, bioinformatics, computational predictions, human genomic databases, and published research limits our ability to identify and understand genetic variables or other potential contributors that impact etiology and phenotypic variability in FXS. Finally, there are limitations in the available research identifying other potentially relevant genes, their variants, and gene–gene–protein interactions, as discussed in our report, indicating that improved understanding of their specific influences on neurodevelopment and CNS functioning is needed to support the development of more effective interventions in FXS.

## 10. Conclusions

Fragile X syndrome often presents a wide range of neurodevelopmental, behavioral, and psychiatric challenges caused by errors in the *FMR1* gene, most often from CGG trinucleotide repeat expansions leading to full mutations by the methylation of the DNA cytosine bases and silencing *FMR1* gene translation, with significantly altered levels or reduced or no production of mRNA and encoded FMRP. The encoded FMR1 protein (FMRP) acts as a synaptic functional regulator that plays a central role in neuronal development, synaptic plasticity, mRNA stability, dendritic transport and protein synthesis including collagen. Despite the significant progress in the knowledge of this complex syndrome, much remains to be understood about the genetic determinants, molecular mechanisms, and biological processes underlying FXS.

Early detection of FXS with methylation specific-quantitative melt analysis (MS-QMA) that targets CpG sites within the *FMR1* intron 1 obtained from DNA in newborn blood has immediate applications in FXS diagnostics, with the potential to be used in nationwide newborn screening. Early detection of FXS allows for intensive developmental therapies and multidisciplinary management, which when instituted early on, improves long-term outcomes. Current treatment strategies are focused on developmental support and symptom management throughout the lifespan, with promising research currently exploring more targeted therapeutic approaches underway, with the potential to reveal improvements in core syndrome deficits. With the goal of improving quality of life, future research and clinical trial efforts should continue to focus on improving early diagnostic practices, developmental supports, and evidence-based treatments.

## Figures and Tables

**Figure 1 genes-16-00149-f001:**
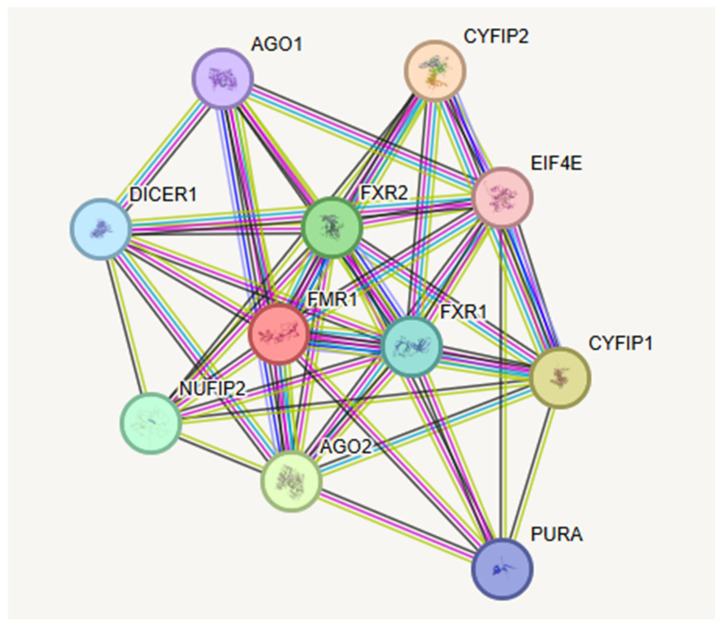
STRING protein–protein interaction network for the *FMR1* gene with functional interactions involving ten associated nodes and 44 edges with their predicted functional interactions including biological processes and molecular functions [28]. Edges represent protein–protein associations which are considered specific and meaningful such as proteins that jointly contribute to a shared function. Network nodes represent proteins with splice isoforms or have post-translational modifications and collapsed into each node for all proteins produced by a single protein-coding gene such as *FMR1* [38].

**Table 1 genes-16-00149-t001:** STRING: predicted functions for *FMR1* Gene *.

*Biological Process (Gene Ontology)*	*Molecular Function (Gene Ontology)*	*Cellular Component (Gene Ontology)*	*KEGG Pathways*	*Reactome Pathways*	*Disease-Gene* *Associations*
Negative regulation of translation	Translation regulator activity	Polysome	RNA transport	Small interfering RNA (siRNA) biogenesis	Fragile X syndrome
Post-transcription regulation of gene expression	siRNA binding	mRNA cap binding complex		MicroRNA (miRNA) biogenesis	Fragile X-associated tremor/ataxia syndrome
RISC assembly	Protein kinase binding	RISC		Regulation of NPAS4 mRNA translation	X-linked monogenic disease
Regulation of translation	RNA 7-methyl-guanosine cap binding	Dendritic filopodium		Competing endogenous RNAs (ceRNAs) regulate translation	
Pre-miRNA processing	Single-stranded RNA binding	Dendritic spine neck		Post-transcription silencing by small RNAs	

* https://www.string-db.org [37].

**Table 2 genes-16-00149-t002:** Predicted functions of top ten significantly associated proteins with FMR1 *.

*Protein Symbol*	*Description*
**CYFIP2**	Cytoplasmic FMR1-interacting protein 2 is involved in T cell adhesion in p53/TP53-dependent induction apoptosis as a component of the WAVE1 complex required for BDNF–NTRK2 endocytic trafficking and signaling from early endosomes
**CYFIP1**	Cytoplasmic FMR1—interacting protein 1 and a component of the CYFIP–EIF4E–FMR1 complex binds to the mRNA cap, mediates translational repression and plays a role in axon overgrowth
**AGO2**	Protein argonaute-2 is required for precise RNA mediated gene silencing by the RNA-induced silencing complex (RISC)
**FXR2**	Fragile X messenger ribonucleoprotein syndrome-related protein 2 is an RNA-binding protein belonging to the FMR1 family
**NUFIP2**	Nuclear fragile X messenger ribonucleoprotein-interacting protein 2 is involved with RNA binding
**FXR1**	Fragile X messenger ribonucleoprotein syndrome-related protein 1 is an RNA-binding protein required for the embryonic and postnatal development of muscle tissue, and it regulates the intracellular transport and local translation of certain mRNAs
**DICER1**	Endoribonuclease Dicer plays a central role in specific post-translational gene silencing events impacting gene function
**PURA**	Transcriptional activator protein Pur-alpha is a transcriptional activator protein that binds to and initiates DNA replication and recombination
**AGO1**	Protein argonaut-1 is required for RNA mediated gene silencing and binds to short RNAs
**EIF4E**	Eukaryotic translation initiation factor 4 E recognizes and binds the 7-methylguanosine-containing mRNA cap during an early step in the initiation of protein synthesis and facilitates ribosome binding

* STRING database (www.string-db.org) [37].

## Data Availability

No new data were created or analyzed in this study. Data sharing is not applicable to this article.

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
