# Peer review of "Systematic Review: Fragile X Syndrome Across the Lifespan with a Focus on Genetics, Neurodevelopmental, Behavioral and Psychiatric Associations"

_genes, 2025, doi:10.3390/genes16020149_

Round 1

Reviewer 1 Report

Comments and Suggestions for Authors

This article by Genovese and Butler comprehensively reviews current information on the genetic and clinical aspects of fragile X syndrome (FXS), including genotypes, systemic and neurodevelopmental features and treatments.

This revies paper is clearly written, comprehensive and informative. However, the reviewer sees several issues, one major and the others minor:

1. Although both the genetic and clinical features are well described, the pathophysiologic link between them remains mostly unexplained. For example, what is the link between the protein-protein interaction (Section 2, Table 1 and Figure 1) and neural connectivity (Section 5)? Additional information about genotype-phenotype correlation and pathophysiology (anatomic, biochemical, physiologic and animal experimental findings) should be provided.

2. Early detection and diagnosis utilizing newborn screening … can improve outcomes for FXS patients (Lines 447-449): Is there any evidence for the usefulness of newborn screening for FXS?

3. There are many font errors: expansion of (Line 125), self-injurious behaviors (Line 286), is (Line 339), Ongoing (Line 373) and ZYN002 (Line 400).

4. There are several typographic errors: influences availability (comma missing, Line 23), H (Line 341), and have been been (Line 249).

Author Response

Comment 1: Although both the genetic and clinical features are well described, the pathophysiologic link between them remains mostly unexplained. For example, what is the link between the protein-protein interaction (Section 2, Table 1 and Figure 1) and neural connectivity (Section 5)? Additional information about genotype-phenotype correlation and pathophysiology (anatomic, biochemical, physiologic and animal experimental findings) should be provided.

Response: Agree. This information has been added.

Comment 2: Early detection and diagnosis utilizing newborn screening … can improve outcomes for FXS patients (Lines 447-449): Is there any evidence for the usefulness of newborn screening for FXS?

Response: Agree. This information has been added.

Comment 3: There are many font errors: expansion of (Line 125), self-injurious behaviors (Line 286), is (Line 339), Ongoing (Line 373) and ZYN002 (Line 400).

Response: Corrections made.

Comment 4: There are several typographic errors: influences availability (comma missing, Line 23), H (Line 341), and have been been (Line 249).

Response: Corrections made.

Reviewer 2 Report

Comments and Suggestions for Authors

Genovese and Butler review the current state on the Fragile X Syndrome focusing on neurodevelopmental, behavioral and psychiatric features across the lifespan.

The manuscript includes no primary data but summarizes literature data addressing the points mentioned. This might be helpful to physicians and care keepers to get a first current overview as well as for patient families as an up date.

Besides from smaller formal points (below) the manuscript would benefit from a summarizing scheme or table including: symptoms, age of onset, frequency and intervention to help the reader through the 15 pages.

Minor points:

-          Add OMIM and ORPHANET numbers for FXS, FXPOI and FXTAS.

-          Line 94: The 2-6% frequency of ASD individuals who have FXS is this number in males or females or both? Be more clear, as this is highly penetrant in males and rare in females!

-          Throughout the text use the abbreviation when first time was introduced e.g. FMRP and do not explain several times.

-          Line 149: genes need to be written italic

-          Table 1 in the current format is not readable.

-          Make a comment if there is a correlation between repeat length and the clinical severeness of symptoms this is only explained for permutation carriers.

-          Line 341: The sentence is not readable.

Author Response

Comment 1: Add OMIM and ORPHANET numbers for FXS, FXPOI and FXTAS.

Response: Agree. Done.

Comment 2: Line 94: The 2-6% frequency of ASD individuals who have FXS is this number in males or females or both? Be more clear, as this is highly penetrant in males and rare in females!

Response: Agree. Clarification added.

Comment 3: Throughout the text use the abbreviation when first time was introduced e.g. FMRP and do not explain several times.

Response: Agree. Edits made.

Comment 4: Line 149: genes need to be written italic.

Response: Agree. Done.

Comment 5: Table 1 in the current format is not readable.

Response: Correction made.

Comment 6: Make a comment if there is a correlation between repeat length and the clinical severeness of symptoms this is only explained for permutation carriers.

Response: Agree. Done.

Comment 7: Line 341: The sentence is not readable.

Response: Sentence has been clarified.